# Elimination of Hepatitis B in Highly Endemic Settings: Lessons Learned in Taiwan and Challenges Ahead

**DOI:** 10.3390/v12080815

**Published:** 2020-07-28

**Authors:** Chun-Jen Liu, Pei-Jer Chen

**Affiliations:** 1Department of Internal Medicine, National Taiwan University Hospital, Taipei 10002, Taiwan; cjliu@ntu.edu.tw; 2Hepatitis Research Center, National Taiwan University Hospital, Taipei 10002, Taiwan; 3Graduate Institute of Clinical Medicine, National Taiwan University College of Medicine, Taipei 10002, Taiwan

**Keywords:** hepatitis B virus, elimination, HBsAg clearance, vaccination, treatment

## Abstract

Hepatitis B virus (HBV) infection and its related liver diseases are important health problems worldwide, particularly in the Asia-Pacific region. For the past 4–5 decades, Taiwan’s government and scientists have cooperated together to control this virus infection and its related liver diseases. These efforts and achievements have made progress toward the elimination of HBV. Taiwan’s government initiated the Viral Hepatitis Control Program (VHCP) in the1970s, and then launched the national vaccination program in 1984. This universal vaccination program effectively decreased the rate of hepatitis B carriage and the development of hepatocellular carcinoma (HCC) in the younger generation. Since 2003, approved anti-HBV treatments were reimbursed nationwide. This reimbursement program resulted in a higher uptake of anti-HBV treatments, which contributed to a decrease in liver-related disease progression and subsequently reduced attributable mortality in Taiwan. This experience can be shared by countries in other parts of the world regarding the control of chronic viral hepatitis B.

## 1. Introduction

Hepatitis B virus (HBV) infection is an important public health problem worldwide. This etiologic agent was not identified until 1965, when Baruch S. Blumberg found the relationship between Australia antigen and serum hepatitis [1,2]. The antigen was later confirmed to be the hepatitis B surface antigen (HBsAg) of the virus. Over the subsequent 20 years, the natural history of HBV infection and routes of transmission were clarified, and an effective vaccine was developed. These achievements led to the effective interruption of transmission in Taiwan and also in other regions of the world [3]. 

End-stage liver diseases (ESLDs) were common in Taiwanese people in the past. An extremely high rate of HBsAg carriage in the general population (15–20%) was noted from epidemiology studies around 1975; about 80% of the patients with hepatocellular carcinoma (HCC) were found to be HBsAg positive [4,5,6,7,8,9,10,11]. Because of this serious health problem, the Taiwanese government organized a Viral Hepatitis Control Committee in 1981 to supervise and coordinate disease control policies. Afterward, the Taiwanese government launched a mass vaccination program against hepatitis B on July 1, 1984; initially focusing on immunization of the babies born to HBsAg carrier mothers, and later on extending to all newborns in 1986. More than thirty years after the implementation of the program, the HBsAg carriage rate in the vaccinated population decreased to ~1% [12,13,14,15,16,17,18,19,20,21]. Most importantly, the development of HCC in the young vaccinees was also found to be reduced in parallel. This is the first time that a human cancer could be prevented by vaccination [22,23]. Later on, the human papillomavirus vaccine was documented as the second successful case to prevent the development of cervical cancer [24]. 

The prevention of new HBV infection was successfully achieved through both vaccination and the effective interruption of the transmission. However, these measures benefit little those with well-established chronic HBV infection. Interferon (IFN)- and nucleos(t)ide analogue (NUC)-based therapies acting against HBV have been developed gradually since 1980s. Although not curative, these anti-HBV treatments improve the clinical outcomes of patients with chronic hepatitis B. Following the recommendation of the Viral Hepatitis Control Committee, the Taiwanese government approved a nationwide reimbursement program for these two regimens for chronic hepatitis B patients in 2003. 

These measures have made progress toward the elimination of HBV in Taiwan [3,25]. We would like to report our experiences with other countries with endemic HBV infection.

## 2. Hepatitis B Vaccination Program in Taiwan

### 2.1. The Universal Hepatitis B Mass Vaccination Program in Infants

Mother-to-infant transmission (MIT) of HBV was found to result in a high rate of chronic HBsAg carriage, and is a major route of HBV transmission in HBV-endemic countries [10,26]. Thus, the most effective way to prevent new HBV infections is to interrupt the virus infection early in life. Beasley et al. [12] and Lo et al. [27] performed pilot studies in newborns and infants, demonstrating that passive and active immunization were effective in interrupting perinatal HBV transmission.

Because the disease burden caused by HBV was big in Taiwan [6,7,8], in 1981, the Taiwanese government developed a Viral Hepatitis Control Committee chaired by Ding-Shinn Chen [13]. Briefly, the National Science Council (now known as the Ministry of Science and Technology) took charge of viral hepatitis research and development, and the Department of Health (now known as the Ministry of Health and Welfare) was responsible for public education and hepatitis B immunization [3]. A mass vaccination program was launched on July 1, 1984 [14], initially targeted at the newborns of hepatitis B carrier mothers and later extending to all newborns in 1986. The coverage rate of vaccination in newborns has generally been >95% for the last 30 years, and recently reached 98% [28].

### 2.2. Effectiveness of Mass Hepatitis B Vaccination in the Decrease in HBsAg Carriage and Liver Diseases

#### 2.2.1. Decrease in HBsAg Carriage in Children and Adolescents

The efficacy of the vaccination program was evaluated 18 months after the launch of the program; 85% of the infants born to HBsAg carrier mothers were protected from chronic HBV infection [15]. To further evaluate the effectiveness of the vaccination program, the Taiwanese experts conducted seroepidemiologic surveys every 5 years after the start of the program. The HBsAg carrier rate in the children of Taipei City decreased from 11% in 1984 [16] to approximately 1% in subsequent surveys, [17,18,19,20] and down to 0.5% in a recent study [21]. Now the decrease in the HBsAg carriage rate in the population has extended to adults around 30 years of age. 

#### 2.2.2. Reduction of Fulminant Hepatitis in Infants and HCC in Childhood

Although the rate of HBsAg carriage in the younger generation has decreased, the most important issue is whether HBV-related ESLDs can also be controled or prevented. Since the start of the vaccination program, mortality related to fulminant hepatitis in infants reduced dramatically, from 5.36 to 1.71 per 100,000 infants [29] and fulminant hepatitis B in children >1 year old has almost disappeared [30]. 

Liver cirrhosis (LC) and HCC are sequelae of long-term HBV infection, usually developing after 4–5 decades of infection [31]. Therefore, the effect of hepatitis B vaccination on the prevention of cirrhosis and HCC in the entire population can only be observed decades after the start of global vaccination. Notably, in Taiwan, HCC is rarely observed in children and all such cases are seropositive for HBsAg [32]. Taking this opportunity, Mei-Hwei Chang and Ding-Shinn Chen followed the incidence of HCC in children in cohorts of 6–9 years of age right before and after the introduction of universal vaccination. They discovered that the incidence of HCC in children 6–9 years of age decreased from 0.52 per 100,000 children in those born before the vaccination program to 0.13 per 100,000 in those born after the program [22]. The decline in HCC was consistently observed in children 6–14 years of age and later in young adults after extended follow-up [24,33]. Recent analyses based on Taiwan HCC registry systems demonstrated that the relative risk (RR) for the development of HCC in the vaccinated verus unvaccinated birth cohort was 0.26 in patients 6–9 years old, 0.34 in patients 10–14 years old, 0.37 in patients 15–19 years old, and 0.42 in patients 20–26 years old [33]. This is the first example that a human cancer (HCC) could be prevented by vaccination against a virus (HBV). Vaccination has been proved to be one of the essential keystones in the eventual elimination of HBV infection [3,22,23,24,25,34,35].

## 3. Nationwide Reimbursement Program for the Treatment of Patients with Chronic Hepatitis B

The vaccine program has been extremely successful at preventing new HBV infection, reducing new HBsAg carriage and HBV-associated mortality in children and young adults in Taiwan. However, the prevalence of HBV infection in the adult population before the era of universal vaccination remains high, with >90% of the adult population greater than 35 years of age having been infected with HBV and 12–15% of them remaining HBsAg positive. 

For those chronically infected with HBV, appropriate management may be needed to reduce the risk of long-term liver sequelae, including LC and HCC. These management methods included (1) identification of individuals at risk; (2) diagnosis and monitoring; and (3) provision of effective treatment. In Taiwan, patients with known chronic HBV infection but without active anti-HBV treatment are followed up at local clinics or hospitals every 6 to 12 months. Liver function tests, alpha-fetoprotein, and abdominal ultrasonography are performed regularly. Subjects without a history of viral hepatitis infection have been able to receive free tests for hepatitis B virus (HBsAg) and hepatitis C virus (anti-HCV) infection at the age of 45 years since 2011. According to our national database, from 2011 to 2018, around 0.7 million patients in total received an HBsAg test, and 14.4–16.1% of them were found to be positive for HBsAg.

### 3.1. Identification of Serum HBV DNA as One of the Key Viral Biomarkers

The identification of factors reliably predicting clinical outcomes of patients with chronic HBV infection is important for clinical physicians. Chien-Jen Chen conducted a large community-based cohort study in Taiwan. In 1991–1992, researchers collected and followed 23,820 Taiwanese subjects from seven townships, including 4155 HBsAg carriers. Several important factors relevant to the development of HBV-related HCC and LC have been clarified from this long-term cohort study [36,37,38,39,40].

In addition to HBsAg and hepatitis B e antigen (HBeAg), Chen‘s team found that (1) the level of serum HBV DNA at recruitment correlated with the risk of HCC [36,37,38,39] and (2) there was a dose–response relationship between the baseline serum HBV DNA level and the development of HCC and LC [38,40]. As for HCC, the cumulative risk was 108 per 100,000 person-years for levels of HBV DNA of <300 copies/mL, which increased to 1152 for those with levels of >10^6^ copies/mL [38]. The cohort study also discovered a similar relationship between HBV viral load and the risk of progression to cirrhosis [40].

Based on these observations, the ultimate goal of HBV treatment is to permanently suppress HBV replication in chronic hepatitis B patients [41,42]. It is critical to initiate treatment in time in those at risk of disease progression. Several effective and safe antiviral agents against HBV have been discovered in the past 2–3 decades. Two main antiviral treatment options exist currently: IFN-alpha and NUCs. IFN is delivered by subcutaneous injection, whereas NUCs are administered orally. Because of the high prevalence of HBV infection in Asia, many of the clinical trials have been conducted and led by Taiwanese investigators [43,44,45,46]. Later on, several large Asian cohorts studies demonstrated the benefits of these antiviral treatments in suppressing the replication of the virus and then reducing the risk of HBV-related morbidities and mortality [44,47]. 

Despite the availability of potent anti-HBV agents and the establishment of Asia-Pacific and Taiwan guidelines for the management of chronic HBV infection [41,42], low percentages of patients actually received treatment in Asia in the 1990s, including in Taiwan. These low treatment rates were due to several factors, such as low awareness amongst carriers of the need for monitoring, insufficient knowledge of physicians regarding treatment status/recommendations, and most importantly, financial barriers to access to treatment.

### 3.2. Providing the Financial Support to Access Treatment

The Viral Hepatitis Control Program (VHCP) in Taiwan played a very important role in the prevention and control of HBV infection. Each stage of the VHCP has had a different focus; universal vaccination being the focus in the early stages of the VHCP and the provision of treatment for chronic viral hepatitis B and C being the aim in 2003. Since then, for patients with chronic hepatitis B indicated for the treatment (as reflected by high serum HBV DNA (≥20,000 IU/mL for HBeAg positive patients and ≥2000 IU/mL for HBeAg negative patients) and a persistently elevated serum ALT (alanine aminotransferase) level (≥2 times upper limit of normal) for non-cirrhotic patients; a high serum HBV DNA level for cirrhotic patients), the national health insurance covers all costs related to outpatient visits, laboratory monitoring, and the drug itself. The costs of both IFN and NUCs are now reimbursed for the treatment of chronic hepatitis B in Taiwan.

### 3.3. Effectiveness of Active Anti-HBV Treatment in the Decrease in All-Cause and Liver-Related Mortalities

The national viral hepatitis therapy program was launched in Taiwan in October 2003. The impact of the program on the reduction of ESLD burden was examined thereafter [48]. In 2011, antiviral therapy was delivered to a total of 157,570 and 61,823 patients with chronic hepatitis B and C, respectively. Registry data demonstrated that incidence rates of ESLDs and mortality declined gradually from 2000–2003 (before the start of the reimbursement program) through 2004–2007 to 2008–2011. These findings were observed in cohorts of various ages and genders. This nationwide database demonstrated that the mortality caused by ESLDs and the incidence and mortality of HCC can be significantly decreased through a nationwide viral hepatitis treatment program. The age-adjusted HCC incidence reached a plateau around 2005, then declined by about 5% a year in both male and female populations (Figure 1). The death toll due to cirrhosis or hepatic failure ranked 6th among the 10 leading causes of death in Taiwan before 2003, but dropped to 7th in 2005, to 8th in 2008, to 9th in 2012, and to 10th in 2015. It is expected to drop out of the list of the 10 leading causes of death in Taiwan in 2020 (Figure 2), and the chart documents the efficacy of our chronic viral hepatitis control program so far. The proportion of HBV-related HCC has decreased gradually over the last 20 years in Taiwan [49]. In 2001, the percentage of HCC attributable to HBV infection was 66% in male patients and 41% in female patients. In 2015, the overall percentage of HBV-related HCC further decreased to 40%. To help demonstrate the contribution of anti-HBV treatment to the reduction of liver-related deaths, we also provide the cumulative number of patients who received reimbursed anti-HBV treatment from 2004 to 2015 in Figure 2.

### 3.4. Approaching the WHO 2030 Goals

The success of the VHCP in the prevention and control of HBV infection to date can be attributed to the Taiwanese government’s determination, well-designed vaccine and treatment reimbursement programs, effective implementation, and appropriate periodic assessment. Furthermore, there are good public health care policies and a clear household registration system, and the Taiwanese are well educated and willing to follow the policies of the government [3]. Very importantly, the nationwide health insurance program was successfully established in 1995 in Taiwan; this system provides medical services accessible to all citizens. All these factors contributed to the success of the HBV control program in Taiwan. 

In 2016, the WHO called for the elimination of viral hepatitis before 2030 [50]. Specifically, at least 90% of new hepatitis viral infections should be prevented, at least 80% of all patients with chronic viral hepatitis indicated for treatment should receive effective anti-viral therapy, and at least 65% of viral hepatitis-related mortality should be prevented through various management methods. The VHCP already proposed a program for the elimination and eradication of viral hepatitis in Taiwan as the ultimate goal in 2015. To achieve the goals proposed by the WHO, several unmet needs to be resolved. 

## 4. Unmet Needs

### 4.1. Improvement of HBV Vaccination Program

Despite these successes in preventing and controlling HBV infection over the last 50 years, there remain challenges regarding the elimination of this virus infection worldwide. Currently, effective measures are available to immunize HBV-naïve subjects, to interrupt the routes of viral transmission, and to treat patients with chronic hepatitis B. However, we still need to resolve several remaining issues in order to achieve the goal of preventing the MIT of HBV [51]: The vaccine coverage rate should be enhanced, an HBV vaccine for outborn neonates should be available, HBV markers should be screened in pregnant women, and highly viremic pregnant mothers should be treated.

Moreover, non-compliance to the vaccination schedule, breakthrough infection, and intrauterine infection are the causes of the vaccine failure [52]. Recent analysis from Taiwan suggested that, among vaccinees, incomplete immunization was the key risk predictor of HCC, fulminant hepatic failure, and chronic liver disease (CLD) [53]. Thus, a continuous commitment to vaccine coverage rate and compliance to the immunization course are important issues to be further resolved.

If the mother is highly viremic/infectious, the MIT of HBV still occurs in approximately 10% of the newborns receiving vaccination [23]. Recent studies identified that serum HBV DNA and quantitative serum HBsAg levels can serve as infectivity markers [54]. For pregnant women with human immunodeficiency virus infection, antiviral intervention during pregnancy can effectively prevent transmission of the virus to the newborn [55]. By analogy, lowering the maternal hepatitis B viral load with antivirals before delivery may also help reduce perinatal HBV transmission from highly viremic mothers. Indeed, several studies have confirmed this possibility [56,57,58]. Chen HL et al. reported that, among 118 HBeAg-positive pregnant women with HBV DNA ≥7.5 log_10_ IU/mL, 62 mothers received tenofovir disoproxil fumarate (TDF) 300 mg daily from 30–32 weeks of gestation until 1 month postpartum [58]. At delivery, the maternal HBV DNA levels of the TDF group were significantly lower (4.29 ± 0.93 log_10_ IU/mL) than those of the 56 control mothers without any medication (8.10 ± 0.56 log_10_ IU/mL, *p* < 0.0001). Of the 121/123 newborns among the TDF group, serum HBV DNA positivity at birth (6.15% versus 31.48%, *p* = 0.0003) and HBsAg positivity at 6 months old (1.54% versus 10.71%, *p* = 0.0481) were also significantly lower than those in the control group. These findings strongly support that treatment with TDF for highly viremic mothers was effective in the prevention of the MIT of HBV. Similar findings were reported by other study groups [57].

Based on these findings, international guidelines currently suggest that for HBsAg carrier mothers with high serum HBV DNA levels (usually >1 million/mL), oral NUC (tenofovir disoproxil furamate or telbivudine) can be administered at the beginning of the third trimester of the pregnancy (gestational age: 27–28 weeks), till one month after delivery [41]. The Taiwanese government already launched antiviral prophylaxis for high viral titer pregnant mothers to further reduce mother-to-infant transmission.

However, the cost of the measurement of serum HBV DNA may be high in some countries. The value of quantitative HBsAg (qHBsAg), a simple and cheap biomarker, as an alternative biomarker for the identification of high risk mothers, is now under active investigation [52]. Besides, in countries with very low HBsAg carriage rates, the cost and benefits of antiviral prophylaxis for the reduction of HBV MIT should be further clarified [59].

### 4.2. Development of Curative Agents/Regimens

Currently available anti-HBV agents can only control, but not eradicate, the virus infection. More effective and even curative anti-HBV agents and strategies are urgently needed. Currently, patients with chronic hepatitis B are treated with IFN or NUCs. The response rate is only approximately 30% for IFN therapy, and the treatment duration of NUC therapy is usually several years or lifelong. Thus, new anti-HBV therapies and strategies should be developed. Currently investigated targets include covalently closed circular HBV DNA, entry inhibitors, capsid modulating agents, the silencing of transcription and translation, and immuno-modulating agents [60]. Promising progress has been made in early phase clinical trials, but truly effective therapies will likely require years to develop.

### 4.3. Control and Preventing Progression into End-Stage Liver Diseases after HBV Seroclearance 

The cure of chronic hepatitis B, as defined by HBsAg seroclearance, is considered a milestone for eliminating virus from the liver. However, the cure of HBV does not mean a cure for underlying liver diseases. As is known, for those chronic hepatitis B (CHB) patients older than 50 years of age or already in the advanced liver fibrosis stage, the clearance of HBV does not completely prevent liver disease progression into liver decompensation or HCC [47]. 

Hepatic fibrosis is a key factor leading to CLD-related morbidity and mortality. To develop effective anti-fibrotic or fibrinolytic therapies, we need to know more about the pathway and the genetic determinants of hepatic fibrosis [61]. The de novo development or recurrence of HCC after HBV curative therapy is another serious challenge; we need to develop new strategies to survey or prevent hepatocarcinogenesis. Because cirrhosis is the most dominant risk factor for HCC, the chemoprevention of hepatocarcinogenesis in cirrhotic patients as a priority population should be investigated.

### 4.4. Control of Liver Diseases and Co-Morbidities Apart from Viral Hepatitis

Finally, the importance of lifestyle changes cannot be overlooked. Several well known co-morbidities, such as diabetes mellitus, non-alcoholic fatty liver disease, and alcohol abuse, accelerate the progress of liver diseases [62]. An active diet and exercise program remains essential for keeping the liver healthy even after being cured of HBV. Certainly, abstinence from alcohol or liver-damaging folk medicines or supplements is advised and should be followed.

## 5. Conclusions

Vaccination and antiviral reimbursement programs effectively control new and chronic HBV infections. The remaining hurdles need to be overcome in order to reach the goal of HBV elimination before 2030. Specifically, the risk of the MIT of HBV should be further minimized; curative regimens for HBV should be developed; cost-effective anti-HBV therapies should be offered globally; effective anti-fibrotic/fibrinolytic agents and chemopreventive regimens for HCC are urgently needed; and lifestyle modifications to control concurrent metabolic liver diseases should be taken as well. 

## Figures and Tables

**Figure 1 viruses-12-00815-f001:**
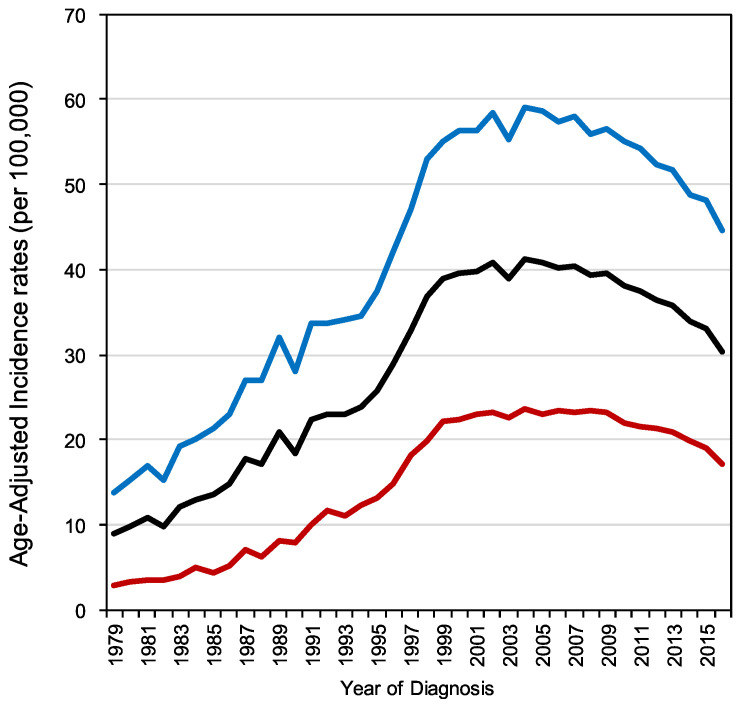
Age-adjusted incidence of hepatocellular carcinoma (HCC) (per 100,000) from 1979 to 2015. Black line: both genders; blue line: male gender; red line: female gender (national reimbursement program for the treatment of chronic viral hepatitis launched in 2003; data released by the Ministry of Health and Welfare, Taiwan).

**Figure 2 viruses-12-00815-f002:**
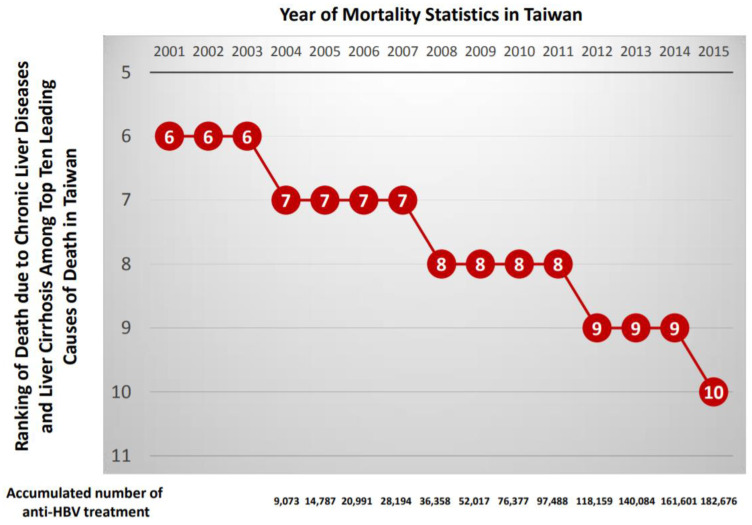
Ranking of deaths due to chronic liver diseases and liver cirrhosis among the top ten leading causes of death in Taiwan (from 2001 to 2015); cumulative number of patients receiving reimbursed anti-hepatitis B virus (HBV) treatment (from 2004 to 2015) is shown at the bottom (national reimbursement program for the treatment of chronic viral hepatitis launched in 2003; data released by the Ministry of Health and Welfare, Taiwan).

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
