# Peer review of "Elimination of Hepatitis B in Highly Endemic Settings: Lessons Learned in Taiwan and Challenges Ahead"

_viruses, 2020, doi:10.3390/v12080815_

Round 1

Reviewer 1 Report

This article titled “Elimination of hepatitis B in highly endemic settings: lessons learned and challenges ahead” reviews the experience Taiwan has had over the past few decades from first implementation infant vaccination programs, to scaling up the vaccination coverage and improving treatment uptake for chronic hepatitis B.  
This short review provides valuable insight into the experience of a hepatitis B endemic country which has aimed to reduce the morbidity and mortality attributable to hepatitis B. I believe this review will be of use for other countries in similar settings.  
A couple of minor suggestions, which I feel will improve the article are as follows:

Line 2: As this review is particularly focused on the experience Taiwan has had I suggest amending the title to reflect this fact. For example: “Elimination of hepatitis B in highly endemic settings: lessons learned in Taiwan and challenges ahead”. This would ensure the reader immediately does not expect a global review.

Line 20: Suggested re-wording of sentence starting with “This reimbursement…” to reflect the fact that the program resulted in higher uptake of anti-HBV treatments (if this is true?) which resulted in a decrease in liver-related disease progression and subsequently reduced attributable mortality in Taiwan.
Otherwise it sounds like the reimbursement caused the reduction in liver related disease, which is not true.

Line 31:  specify in what regions these achievements led to interruption of transmission – was that only in Taiwan? Then state this or was it more generally ‘in some regions’ ?

Line 42:  This sentence is not true. The Human Papillomavirus (HPV) vaccine prevents against some forms of cancer such as cervical cancer. I suggest re-phrasing sentence to make your point. This comment also applies to the sentence on line 97/98.

Line 59:  Insert reference right after mention of authors not at the end of the sentence to make it easier for the reader to follow.

Line 68:  Do the authors have a reference they can cite for the vaccination coverage reaching 98% ?

Line 109:  Can the authors provide more information or comment on the current/historical surveillance system in Taiwan? e.g. what proportion of people living with chronic hepatitis B have been identified or diagnosed? and what proportion are monitored regularly?

Line 172:  Interesting Figure (2) – how much of the death due to chronic liver disease is attributable to chronic hepatitis B over the timeline of this figure? Is that known?

Author Response

Comments:

This article titled “Elimination of hepatitis B in highly endemic settings: lessons learned and challenges ahead” reviews the experience Taiwan has had over the past few decades from first implementation infant vaccination programs, to scaling up the vaccination coverage and improving treatment uptake for chronic hepatitis B.

This short review provides valuable insight into the experience of a hepatitis B endemic country which has aimed to reduce the morbidity and mortality attributable to hepatitis B. I believe this review will be of use for other countries in similar settings.

A couple of minor suggestions, which I feel will improve the article are as follows:

Line 2: As this review is particularly focused on the experience Taiwan has had I suggest amending the title to reflect this fact. For example: “Elimination of hepatitis B in highly endemic settings: lessons learned in Taiwan and challenges ahead”. This would ensure the reader immediately does not expect a global review.

Response: revised as requested (line 3)

Line 20: Suggested re-wording of sentence starting with “This reimbursement…” to reflect the fact that the program resulted in higher uptake of anti-HBV treatments (if this is true?) which resulted in a decrease in liver-related disease progression and subsequently reduced attributable mortality in Taiwan.

Otherwise it sounds like the reimbursement caused the reduction in liver related disease, which is not true.

Response: Thanks for the suggestion; we modified the sentence to make our expression more clear (line 22).

Line 31: specify in what regions these achievements led to interruption of transmission – was that only in Taiwan? Then state this or was it more generally ‘in some regions’ ?

Response: Thanks for the suggestion; we modified the sentence to make our expression more clear (line 34).

Line 42: This sentence is not true. The Human Papillomavirus (HPV) vaccine prevents against some forms of cancer such as cervical cancer. I suggest re-phrasing sentence to make your point. This comment also applies to the sentence on line 97/98.

Response: Thanks for the query, we edited our sentence and added relevant part on HPV vaccine. Actually, the HPV vaccine program was initiated in 2001 (Harper DM et al, Lancet 2006;367:1247) and its impact on the reduction of cervical cancer was published in 2007 (Garland SM et al, New Engl J Med 2007;356:1928), about ten years after the observation that HBV vaccination decreased the development of HCC. Thus we believed that similar prevention of cancer development can be achieved through HPV vaccination; but this benefit was documented right after that of HBV vaccine regarding the prevention of HCC. (revised reference 28)

Line 59: Insert reference right after mention of authors not at the end of the sentence to make it easier for the reader to follow.

Response: revised as requested (line 64)

Line 68: Do the authors have a reference they can cite for the vaccination coverage reaching 98% ?

Response: We provided the news and data released by the Taiwan government in year 2017 (in Chinese). (revised reference 28)

https://www.mohw.gov.tw/dl-48435-4f5e5141-5b04-4e49-8f96- 0642ec0771c9.html

Line 109: Can the authors provide more information or comment on the current/historical surveillance system in Taiwan? e.g. what proportion of people living with chronic hepatitis B have been identified or diagnosed? and what proportion are monitored regularly?

Response: In Taiwan, patients with known chronic HBV infection without active anti-HBV treatment are followed at local clinic or hospital every 6 to 12 months. Liver function test, alfa-fetoprotein and abdominal ultrasonography will be performed regularly. For subjects without information about viral hepatitis infection status, they can receive free test for hepatitis B virus (HBsAg) and hepatitis C virus (anti-HCV) infection at the age of 45 years since year 2011. According to our national database, from 2011 to 2018, totally around 0.7 million patients received HBsAg test, and 14.4~16.1% of them were found to be positive for HBsAg. Unfortunately, we do not have the exact number or proportion of people living with HBV infection who are identified and monitored regularly. We added this information in the text part (line 118).

Line 172: Interesting Figure (2) – how much of the death due to chronic liver disease is attributable to chronic hepatitis B over the timeline of this figure? Is that known?

Response: We do not have the exact number as the mortality data did not record exact etiology. However, we did provide the changing proportion of HBV in HCC in the last 20 years [49]. The proportion of HBV-related HCC decreased gradually [49]. In 2001, the percentage of HCC attributable to HBV infection was 66% in male patients, and 41% in female patients. In 2015, the overall percentage of HBV-related HCC further decreased to 40%. To help demonstrate the contribution of anti-HBV treatment to the reduction of liver-related deaths, we also provided the accumulative number of patients who received anti-HBV treatment from year 2004 to year 2015 in Figure 2. Furthermore, to help demonstrate the contribution of anti-HBV treatment to the reduction of liver-related deaths, in Figure 2, we provided the accumulative number of patients who received anti-HBV treatment from year 2004 to year 2015. (revised Figure 2 and line 179)

Reviewer 2 Report

Manuscript ID: viruses-843575

Elimination of Hepatitis B in Highly Endemic Settings: Lessons Learned and Challenges Ahead

HBV infection and its related liver diseases are important public health problems worldwide. To overcome these problems, Taiwan’s government launched the Viral Hepatitis Control and Program (VHCP) since 1970's and vaccination program since 1984. Furthermore, reimbursement program for anti-HBV treatments has been contributed to reduce liver disease progression and mortality in Taiwan.

In this paper, the authors reviewed these government approaches such as vaccination program and reimbursement program, and discussed unsolved problems. However, I have several concerns that need to be addressed before considering publication. Thus, several points as indicated below need to be addressed by authors to improve the quality of the article.

Major comments

Comment 1: The title of this paper is “ Elimination of Hepatitis B in Highly Endemic settings: Lessons Learned and Challenges Ahead”. However, I think there is a big gap between the title and the information obtained from the contents of this paper. We know that the vaccination and the treatment of anti-HBV agents are effective for preventing viral infection/transmission and reducing HBV DNA of infected patients. Then, what is the most strength point of successful elimination of HBV in Taiwan? Is that reimbursement program? If so, I would think this paper is suitable for another journal, not in the Viruses. The authors need to reconsider focal point of this paper.

Comment 2: (Line 94-97)the relative risk (RR) for the development of HCC in the vaccinated cohort was 0.26 in patients 6-9 years old, 0.34 in patients 10-14 years old, 0.37 in patients 15-19 years old, and 0.42 in patients 20-26 years old in comparison to the unvaccinated cohort [33].

I found authors’ explanation might lead misunderstanding. The authors might need to reconsider explanation, because ref 33 explained these values as “the HCC incidence rate ratios for the vaccinated/unvaccinated “birth” cohorts.

Comment 3: (Line 97-98) This is the first time that a human cancer was prevented by vaccination against a virus.

This sentence should be reconsidered. Does this sentence mean “This (ref33) is the first article reporting that a human cancer was prevented by vaccination against a virus.”? If so, it’s not correct.

Comment 4: (Section 3) In this section, the authors summarized nationwide reimbursement program in Taiwan. However, the subsection content does not match the section title. The authors should reconsider the structure of text and contents of section. Additionally, the title of subsection 3.2 does not match the contents.

Comment 5 (Figure 2): There is very little information from this graph. However, the readers may have an interest the relationship between these data and the contribution of chronic viral hepatitis control program. Therefore, to show more clearer the effectiveness of control program, I suggest adding information such as the number of people who have been vaccinated, who have been treated with anti-HBV agents, and so on in this graph, if it’s possible.

Minor comments

Comment 6: Please doublecheck careless mis-spelling, such as “NATIONAWIDE (NATIONWIDE)” (Line 100), “seerum (serum)” (Line120), “agenta (agents)” (Line134), “monitoing (monitoring)” (Line 149), “monthe (month)” (Line 224), “absteinece (abstinence), “modluating (modulating)” (Line 238),  …

Several abbreviations such as CLD(Line 203) ,TDF (Line 213) CHB (Line 244), qHBsAg (quantitative HBsAg) (Line 228)… should be defined at first mention.

Comment 7: (Line 17) First mention of Viral Hepatitis Control Program should be added "(VHCP)" . Because VHCP is suddenly described on line 142. Similarly, Liver cirrhosis (Line 84) should be added “(LC)”.

Comment 8: (Line 185) "The WHO in 2016 called for elimination of viral hepatitis before 2030." Reference should be added.

Comment 9: Reference numbers are duplicated.

Author Response

Comments:

HBV infection and its related liver diseases are important public health problems worldwide. To overcome these problems, Taiwan’s government launched the Viral Hepatitis Control and Program (VHCP) since 1970's and vaccination program since 1984. Furthermore, reimbursement program for anti-HBV treatments has been contributed to reduce liver disease progression and mortality in Taiwan.

In this paper, the authors reviewed these government approaches such as vaccination program and reimbursement program, and discussed unsolved problems. However, I have several concerns that need to be addressed before considering publication. Thus, several points as indicated below need to be addressed by authors to improve the quality of the article.

Comment 1: The title of this paper is “Elimination of Hepatitis B in Highly Endemic settings: Lessons Learned and Challenges Ahead”. However, I think there is a big gap between the title and the information obtained from the contents of this paper. We know that the vaccination and the treatment of anti-HBV agents are effective for preventing viral infection/transmission and reducing HBV DNA of infected patients. Then, what is the most strength point of successful elimination of HBV in Taiwan? Is that reimbursement program? If so, I would think this paper is suitable for another journal, not in the Viruses. The authors need to reconsider focal point of this paper.

Response: We do not agree that there is a big gap between the title and the information obtained from the contents of this paper. We believe that both the vaccination program and the treatment of anti-HBV agents through nationwide reimbursement program are important and effective for preventing viral infection/transmission and reducing HBV-related morbidity and mortality. And both programs/strategies are critical steps towards the elimination of hepatitis B in highly endemic countries. Thus we emphasize both parts in our manuscript. We believe that these experiences can help other countries of similar HBV infection burden.

Comment 2: (Line 94-97) “the relative risk (RR) for the development of HCC in the vaccinated cohort was 0.26 in patients 6-9 years old, 0.34 in patients 10-14 years old, 0.37 in patients 15-19 years old, and 0.42 in patients 20-26 years old in comparison to the unvaccinated cohort [33].”

I found authors’ explanation might lead misunderstanding. The authors might need to reconsider explanation, because ref 33 explained these values as “the HCC incidence rate ratios for the vaccinated/unvaccinated “birth” cohorts.

Response: Thanks for the query. According to the abstract of the paper, the authors indeed mentioned that “The RRs for HCC in patients 6-9 years old, 10-14 years old, 15-19 years old, and 20-26 years old who were vaccinated vs unvaccinated were 0.26 (95% confidence interval [CI], 0.17-0.40), 0.34 (95% CI, 0.25-0.48), 0.37 (95% CI, 0.25-0.51), and 0.42 (95% CI, 0.32-0.56), respectively.” Nevertheless, we edited the sentence to make the expression more clear. (line 102)

Comment 3: (Line 97-98) “This is the first time that a human cancer was prevented by vaccination against a virus.”

This sentence should be reconsidered. Does this sentence mean “This (ref33) is the first article reporting that a human cancer was prevented by vaccination against a virus.”? If so, it’s not correct.

Response: This is the first example that a human cancer (HCC) can be prevented by vaccination against a virus (HBV) (Chang MH. Hepatitis B virus and cancer prevention. Recent Results Cancer Res 2011;188:75-84). We do not mean that the reference 33 is the first article reporting of this finding. Similar experience was noted later on regarding human papillomavirus virus (HPV) vaccination and the development of cervical cancer. The HPV vaccine program was initiated in 2001 and its impact on the reduction of cervical cancer was documented in 2007; about 10 years after the observations in Taiwan. (line 47)

Comment 4: (Section 3) In this section, the authors summarized nationwide reimbursement program in Taiwan. However, the subsection content does not match the section title. The authors should reconsider the structure of text and contents of section. Additionally, the title of subsection 3.2 does not match the contents.

Response: In this session, we described how the Taiwan government overcame the financial barrier for the treatment of chronic viral hepatitis B and C through providing nationwide reimbursement cost. Thus we believe that content supported the subsection title. However, to more accurately match the content, we modified the title of the subsection. (line 155)

Comment 5 (Figure 2): There is very little information from this graph. However, the readers may have an interest the relationship between these data and the contribution of chronic viral hepatitis control program. Therefore, to show more clearer the effectiveness of control program, I suggest adding information such as the number of people who have been vaccinated, who have been treated with anti-HBV agents, and so on in this graph, if it’s possible.

Response: We believed that the decrease in the number of deaths due to chronic liver diseases was mainly due to the nationwide reimbursement program for the treatment of chronic viral hepatitis B and C. Thus in the revision, we only provided the data regarding the number of patients who had been treated with anti-HBV agents. Detailed information regarding the number of patients who had been treated for HBV infection from year 2004 to year 2011 had already been provided in subsection 3.3. In Figure 2, we modified the content by adding the accumulative number of patients who received anti-HBV treatment from year 2004 to year 2015. (revised figure 2)

Minor comments:

Comment 6: Please doublecheck careless mis-spelling, such as “NATIONAWIDE (NATIONWIDE)” (Line 100), “seerum (serum)” (Line120), “agenta (agents)” (Line134), “monitoing (monitoring)” (Line 149), “monthe (month)” (Line 224), “absteinece (abstinence), “modluating (modulating)” (Line 238), …

Response: we correct these typos.

Several abbreviations such as CLD (Line 203) ,TDF (Line 213) CHB (Line 244), qHBsAg (quantitative HBsAg) (Line 228)… should be defined at first mention.

Response: we provide the full name.

Comment 7: (Line 17) First mention of Viral Hepatitis Control Program should be added "(VHCP)" . Because VHCP is suddenly described on line 142. Similarly, Liver cirrhosis (Line 84) should be added “(LC)”.

Response: we provide the full name.

Comment 8: (Line 185) "The WHO in 2016 called for elimination of viral hepatitis before 2030." Reference should be added.

Response: we add the reference (revised reference 49).

www.who.int/hepatitis/publications/hep-elimination-by-2030

Comment 9: Reference numbers are duplicated.

Response: we correct the duplicate, which is due to technical problem of the editorial transformation system.

Round 2

Reviewer 2 Report

I have checked your revisions and the revised manuscript should be improved yet.

Comment 3: (Line 97-98) “This is the first time that a human cancer was prevented by vaccination against a virus.”

This sentence should be reconsidered. Does this sentence mean “This (ref33) is the first article reporting that a human cancer was prevented by vaccination against a virus.”? If so, it’s not correct.

Response: This is the first example that a human cancer (HCC) can be prevented by vaccination against a virus (HBV) (Chang MH. Hepatitis B virus and cancer prevention. Recent Results Cancer Res 2011;188:75-84). We do not mean that the reference 33 is the first article reporting of this finding. Similar experience was noted later on regarding human papillomavirus virus (HPV) vaccination and the development of cervical cancer. The HPV vaccine program was initiated in 2001 and its impact on the reduction of cervical cancer was documented in 2007; about 10 years after the observations in Taiwan. (line 47)

Comment to the response: It’s not improved yet.

“This” is the first time that…

To avoid misleading, “This” should be reconsidered to another expression or remove this sentence from here.

Comment 5 (Figure 2): There is very little information from this graph. However, the readers may have an interest the relationship between these data and the contribution of chronic viral hepatitis control program. Therefore, to show more clearer the effectiveness of control program, I suggest adding information such as the number of people who have been vaccinated, who have been treated with anti-HBV agents, and so on in this graph, if it’s possible.

Response: We believed that the decrease in the number of deaths due to chronic liver diseases was mainly due to the nationwide reimbursement program for the treatment of chronic viral hepatitis B and C. Thus in the revision, we only provided the data regarding the number of patients who had been treated with anti-HBV agents. Detailed information regarding the number of patients who had been treated for HBV infection from year 2004 to year 2011 had already been provided in subsection 3.3. In Figure 2, we modified the content by adding the accumulative number of patients who received anti-HBV treatment from year 2004 to year 2015. (revised figure 2)

Comment to the response: Figure 2 is not replaced by new version in the revised manuscript. Please confirm it.

Another point,

In the manuscript text (line 182-183) “… we also provided the accumulative number of patients who received anti-HBV treatment from…”

In the figure 2 legend (line 191-192) “ …accumulative number of patients receiving reimbursed anti-HBV treatment…”

Which is the correct expression “the patient receiving anti-HBV treatment” or  “the patient receiving reimbursed anti-HBV treatment” ?

Author Response

Comment 3: (Line 97-98) “This is the first time that a human cancer was prevented by vaccination against a virus.”

This sentence should be reconsidered. Does this sentence mean “This (ref33) is the first article reporting that a human cancer was prevented by vaccination against a virus.”? If so, it’s not correct.

Response: This is the first example that a human cancer (HCC) can be prevented by vaccination against a virus (HBV) (Chang MH. Hepatitis B virus and cancer prevention. Recent Results Cancer Res 2011;188:75-84). We do not mean that the reference 33 is the first article reporting of this finding. Similar experience was noted later on regarding human papillomavirus virus (HPV) vaccination and the development of cervical cancer. The HPV vaccine program was initiated in 2001 and its impact on the reduction of cervical cancer was documented in 2007; about 10 years after the observations in Taiwan. (line 47)

Comment to the response: It’s not improved yet.

“This” is the first time that…

To avoid misleading, “This” should be reconsidered to another expression or remove this sentence from here.

Responses: We modified the sentence as suggested. (line 110)

Comment 5 (Figure 2): There is very little information from this graph. However, the readers may have an interest the relationship between these data and the contribution of chronic viral hepatitis control program. Therefore, to show more clearer the effectiveness of control program, I suggest adding information such as the number of people who have been vaccinated, who have been treated with anti-HBV agents, and so on in this graph, if it’s possible.

Response: We believed that the decrease in the number of deaths due to chronic liver diseases was mainly due to the nationwide reimbursement program for the treatment of chronic viral hepatitis B and C. Thus in the revision, we only provided the data regarding the number of patients who had been treated with anti-HBV agents. Detailed information regarding the number of patients who had been treated for HBV infection from year 2004 to year 2011 had already been provided in subsection 3.3. In Figure 2, we modified the content by adding the accumulative number of patients who received anti-HBV treatment from year 2004 to year 2015. (revised figure 2)

Comment to the response: Figure 2 is not replaced by new version in the revised manuscript. Please confirm it.

Response: We have already uploaded the revised Figure 2 in last revision process. Please check the submitted files in last revision. I upload the revised Figure 2 in this revision also.

Another point,

In the manuscript text (line 182-183) “… we also provided the accumulative number of patients who received anti-HBV treatment from…”

In the figure 2 legend (line 191-192) “ …accumulative number of patients receiving reimbursed anti-HBV treatment…”

Which is the correct expression “the patient receiving anti-HBV treatment” or “the patient receiving reimbursed anti-HBV treatment” ?

Response: reimbursed anti-HBV treatment. (line 196)